# The PagKNAT2/6b-PagBOP1/2a Regulatory Module Controls Leaf Morphogenesis in *Populus*

**DOI:** 10.3390/ijms23105581

**Published:** 2022-05-17

**Authors:** Yanqiu Zhao, Yifan Zhang, Weilin Zhang, Yangxin Shi, Cheng Jiang, Xueqin Song, Gerald A. Tuskan, Wei Zeng, Jin Zhang, Mengzhu Lu

**Affiliations:** 1State Key Laboratory of Subtropical Silviculture, Zhejiang A&F University, Hangzhou 311300, China; yanqiusunny@163.com (Y.Z.); 2019602042066@stu.zafu.edu.cn (Y.Z.); 2020102062014@stu.zafu.edu.cn (W.Z.); 13280827930@163.com (Y.S.); jiangc@zafu.edu.cn (C.J.); zengw@zafu.edu.cn (W.Z.); 2State Key Laboratory of Tree Genetics and Breeding, Key Laboratory of Tree Breeding and Cultivation of the National Forestry and Grassland Administration, Research Institute of Forestry, Chinese Academy of Forestry, Beijing 100091, China; xqsong@caf.ac.cn; 3Center for Bioenergy Innovation, Biosciences Division, Oak Ridge National Laboratory, Oak Ridge, TN 37830, USA; tuskanga@ornl.gov

**Keywords:** *Populus*, leaf morphogenesis, *PagKNAT2/6b*, *PagBOP1/2a*

## Abstract

Leaf morphogenesis requires precise regulation of gene expression to achieve organ separation and flat-leaf form. The poplar KNOTTED-like homeobox gene *PagKNAT2/6b* could change plant architecture, especially leaf shape, in response to drought stress. However, its regulatory mechanism in leaf development remains unclear. In this work, gene expression analyses of *PagKNAT2/6b* suggested that *PagKNAT2/6b* was highly expressed during leaf development. Moreover, the leaf shape changes along the adaxial-abaxial, medial-lateral, and proximal-distal axes caused by the mis-expression of *PagKNAT2/6b* demonstrated that its overexpression (*PagKNAT2/6b* OE) and SRDX dominant repression (*PagKNAT2/6b* SRDX) poplars had an impact on the leaf axial development. The crinkle leaf of *PagKNAT2/6b* OE was consistent with the differential expression gene *PagBOP1/2a* (*BLADE-ON-PETIOLE*), which was the critical gene for regulating leaf development. Further study showed that *PagBOP1/2a* was directly activated by PagKNAT2/6b through a novel *cis*-acting element “CTCTT”. Together, the PagKNAT2/6b-PagBOP1/2a module regulates poplar leaf morphology by affecting axial development, which provides insights aimed at leaf shape modification for further improving the drought tolerance of woody plants.

## 1. Introduction

The leaf morphology not only reflects the evolution of each lineage characterized by photosynthesis but also partially determines the response of plants to environmental stimuli [1]. Several studies on the leaf shape of woody plants, such as grape [2] and poplar [3], suggest that leaf size and structure are related to paleoclimate. Branch length and leaf size contribute to the diversity of canopy architecture, and the small leaf size and compact canopy make forest trees more drought-tolerant [4]. As a woody model species, *Populus* are widely distributed in the northern hemisphere with diverse habitats. Therefore, variation in leaf shape and size may determine their adaptation and coping with global climate change.

The leaves originate from the shoot apical meristem (SAM) and undergo sequential coordinated cell proliferation and expansion along multiply axes which determine their final size and shape [5]. The polarity of leaf development is achieved in three directions: adaxial-abaxial axis (top-to-bottom), proximal-distal axis (base-to-tip), and subsequent medial-lateral axis (center-to-edge) [6,7]. These axial developments are regulated by transcriptional regulators, phytohormones, miRNAs, and small regulatory peptides [5]. Several known developmental regulators include Class I *KNOTTED1-like homeobox* (*KNOX I*) genes [8,9], *BLADE ON PETIOLE* (*BOP*) [10], *ASYMMETRIC LEAVES 1* (*AS1*), *AS2* [6,11], *YABBY* [12], and others [13].

The *KNOX I* gene *KNOTTED 1* (*KN1*), which was first identified by a series of dominant mutations affecting leaf development in maize (*Zea mays*), was named for the knotted shape of its leaves [14,15]. Four members of Class I *KNOX* in *Arabidopsis*, including *SHOOT MERISTEMLESS* (*STM*), *BREVIPEDICELLUS* (*BP*, also named *KNAT1*), *KN1*-Like in *Arabidopsis thaliana* 2 (*KNAT2*), and *KNAT6*, play critical roles in establishing and maintaining SAM activity and organ separation during vegetative growth [9]. Numerous studies have confirmed, also in other species, their functions in SAM formation [16,17,18] and lateral organ separation [8,9]. In addition, three *KNOX I* genes are also involved in wood formation and the adaxial xylem cell differentiation [19,20,21]. The homologs of REVOLUTA (REV), PHABULOSA (PHB), and YABBY in maize are suggested to coordinate the regulation of leaf organ adaxial/abaxial polarity, based on the *cis*-acting element “TGAC” in their promoters that were bound by KN1 [22,23]. On the other hand, the leaf of *bop1 bop2* (*BLADE-ON-PETIOLE 1* and *2*) double mutants display a range of developmental defects [24], whereas *BOP1* or *BOP2* overexpressing plants exhibit short and compact architecture, hyponastic and inward curving leaves [25]. In addition, *BOP1* and *BOP2*, which are lateral organ boundary genes, not only promote morphological asymmetry during leaf development but also affect floral patterning, indicating that *BOP1* and *BOP2* are key regulators required for cells differentiation into lateral organs by determining their fate and polarity [10,24,26]. However, the detailed interaction of the aforementioned regulators on leaf morphogenesis is still elusive.

In our previous study, we have characterized *PagKNAT2/6b* in hybrid poplar (*Populus alba × P. glandulosa*) ‘84K’, which makes a rapid and long-lasting response to drought stress [27]. Its overexpression lines exhibit plant architectures similar to plants grown under drought stress, such as dwarf plants with small leaves. Thus, we hypothesized that *PagKNAT2/6b* might affect leaf morphogenesis by regulating the expression of essential genes related to leaf development. Here, the molecular mechanism of *PagKNAT2/6b* affecting leaf development is investigated, which not only explores the regulatory mechanism of poplar leaf shape development but also provides a basis for further improving the drought tolerance of woody plants by adjusting the leaf shape.

## 2. Results

### 2.1. PagKNAT2/6b Is Expressed Broadly in Poplar Leaves

To understand the biological function of *PagKNAT2/6b* in poplar leaf development, we firstly evaluated the expression of *PagKNAT2/6b* in leaves. The leaf plastochron index (LPI) measured leaves at different developmental stages [28] and the first leaf separated from the SAM as LPI1. In hybrid poplar ‘84K’, leaves typically achieve full expansion between LPI3 and LPI4 of one-month-old seedlings. The gene expression of *PagKNAT2/6b* increased 5-fold from LPI1 to LPI3 and maintained high expression in LPI3 and LPI4 (Figure 1A). Moreover, β-galactosidase (GUS) staining of *P_PagKNAT2/6b_::GUS* transgenic plants indicated that *PagKNAT2/6b* is expressed throughout the whole process of leaf development (Figure 1B). Meanwhile, the GUS signals in SAM, later leaf primordia, and the boundary between unexpanded leaves were histologically observed (Figure 1C,D), which suggested that *PagKNAT2/6b* might play a key role in leaf initiation and separation. The polar development of leaves is crucial to ensure morphological completion and is achieved mainly through three directions: the adaxial-abaxial axis, the proximal-distal axis, and the medial-lateral axis [6,7]. In LPI1, strong GUS signals were detected in the cells on the adaxial side along the adaxial-abaxial axis and leaf margin cells along the medial-lateral axis in the unexpanded leaf (Figure 1E). In the main leaf midrib of expanded leaf LPI2, the GUS signals were mainly concentrated in the xylem and phloem tissue (Figure 1F) and speculated that these might be involved in regulating adaxial-abaxial axis differentiation in leaf vascular tissues. Furthermore, an intensive GUS signal was detected in the proximal position of the proximal-distal axis of LPI2 (Figure 1B), and on the cells near the main leaf midrib of the medial-lateral axis of LPI3 via histological observation (Figure 1G). Taken together, the localization of *PagKNAT2/6b* on different axes suggested that it may be involved in the leaf initiation and expansion process in three developmental axes.

### 2.2. Phenotypic Changes of Transgenic Poplars Are Caused by Abnormal Expression of PagKNAT2/6b

Leaf development was affected by overexpression and dominant repression of *Pag**KNAT2/6b* in poplar. To further explore the regulatory role of *Pag**KNAT2/6b* in leaf development, a detailed leaf morphological characterization of two-month-old *Pag**KNAT2/6b* overexpression lines (OE5 and OE7) and *PagKNAT2/6b* SRDX dominant repression lines (SRDX1 and SRDX20) was performed (Figure 2A). The mature leaves (LPI3) of *PagKNAT2/6b*-OE and *PagKNAT2/6b*-SRDX lines displayed significant leaf morphology characteristics (Figure 2B). LPI3 of OE lines displayed small, compact leaves, almost no petiole, and inward curled leaves with deeply lobed serrations on leaf margins (Figure 2C,D). In contrast, LPI3 of SRDX lines showed slender and flat leaves with a smooth margin which suggested that *PagKNAT2/6b* could mainly impact the leaf aspect ratio and margin by regulating the development of the proximal-distal axis and medial-lateral axis.

To evaluate the development of poplar leaves in the proximal-distal axis and medial-lateral axis, we measured the leaf length and width and calculated the length/width ratio in control and *PagKNAT2/6b* transgenic lines. Compared with the control plants, the leaf length and width of the OE and SRDX lines were significantly reduced (Figure 2E,F). However, the length/width ratio of OE and SRDX lines showed opposite trends—the length/width ratios of OE5 and OE7 were significantly reduced by 31% and 75%, but that of SRDX1 and SRDX20 increased by 15% and 27%, respectively (Figure 2G). These results indicate that *PagKNAT2/6b* regulates the development of poplar leaves in both the proximal-distal axis and medial-lateral axis.

On the adaxial-abaxial axis compared with the control plants (Figure 3A), the cell number decreased on the adaxial side of the main leaf midrib and increased on the abaxial side of the lateral leaf midrib in OE lines (Figure 3B,C). In contrast, the cell number only decreased on the abaxial side of the lateral leaf midrib in SRDX lines (Figure 3D,E). This suggests that the inwardly curled leaves of OE lines may be due to the asymmetric differentiation of the cells along the adaxial and abaxial sides, and the chief function of *PagKNAT2/6b* may be to regulate the asymmetric differentiation of the lateral leaf midrib based on the SRDX phenotype.

### 2.3. Key Genes for Leaf Development Are Affected in PagKNAT2/6b Transgenic Plants

As overexpression of *PagKNAT2/6b* altered the leaf shape, we analyzed the gene expression profiles during the leaf expansion stage (from LPI2 to LPI4) in two-month-old *PagKNAT2/6b* transgenic and control lines using RNA-Seq. Compared with the control plants, 1999 and 1051 genes were up-regulated and 1664 and 733 genes were down-regulated in OE5 and OE7, respectively (Appendix A). In the SRDX lines, 60 up- and 68 down-regulated genes were identified in SRDX1, but 945 up- and 1316 down-regulated genes were identified in SRDX20, respectively (Appendix A). The difference in the number of differentially expressed genes (DEGs) between the two SRDX lines might be caused by the difference in the expression levels of *PagKNAT2/6b-SRDX*. A total of 144 common-DEGs were identified (Appendix A), which were defined as the opposite expression patterns in the OE and SRDX lines, i.e., 107 genes were up-regulated in OE5 or OE7 but down-regulated in SRDX1 or SRDX20, and 37 genes were down-regulated in OE5 or OE7 but up-regulated in SRDX1 or SRDX20 (Figure 4A). Among the common-DEGs, ten known leaf development-related genes were selected as key-DEGs (Appendix A) and further validated by qRT-PCR, and whose expressions were consistent with that of the RNA-Seq (Figure 4B).

To validate the potential function of the common-DEGs, we generated a 1-hop co-expression network centered on *PagKANT2/6b* and the common-DEGs based on the *Populus* Gene Atlas database covering various tissues and organs (Appendix A). The GO enrichment analysis of genes in the co-expression network revealed that the common-DEGs affected by the mis-expression of *PagKNAT2/6b* were associated with the biological processes of “response to oxidative/biotic/chemical stresses” and “cell wall organization/modification” (Appendix A). These results indicate that poplars overexpressing *PagKNAT2/6b* mimic the changes in leaf morphogenesis in response to stress, and these changes are also associated with cell wall organization.

### 2.4. PagKNAT2/6b Directly Activates PagBOP1/2a Expression

Among the key-DEGs, *PagBOP1/2a* and *PagBOP1/2b*, orthologs of leaf polarity developmental genes *BOP1* and *BOP2* in *Arabidopsis* [29], were highly induced in *PagKNAT2/6b*-OE lines and repressed in SRDX line (Appendix A). The leaf morphology of *PagKNAT2/6b* overexpressing poplars was similar to *BOP1* and *BOP2* overexpressing *Arabidopsis* [25]. The expression of *PagBOP1/2a* in the leaf of *PagKNAT2/6b* OE and SRDX significantly increased and decreased than that of *PagBOP1/2b*. Moreover, the expression of *PagBOP1/2a* was rapidly induced by drought stress, which is similar to *PagKNAT2/6b* (Appendix A). Therefore, we tested the transactivation activity of *PagKNAT2/6b* to the promoter of *PagBOP1/2a* using a transient expression assay. As shown in Figure 5A, the promoter of *PagBOP1/2a* was highly activated by PagKNAT2/6b. To further investigate whether the activation of *PagBOP1/2a* by PagKNAT2/6b was dependent on the previously identified *cis*-acting element “TGAC” of its homolog gene KN1 in maize, we divided the 500-bp *PagBOP1/2a* promoter region (Pro) into two fragments (“Pro1” containing two “TGAC” elements and “Pro2” without “TGAC”) and compared the activation of PagKNAT2/6b to the fragments using Y1H assay. Interestingly, PagKNAT2/6b could bind to the “Pro2” without “TGAC” of *PagBOP1/2a* promoter but could not bind to the “Pro1” containing two “TGAC” elements (Figure 5B). This result suggested that PagKNAT2/6b might bind to the promoter of *PagBOP1/2a* through other *cis*-acting elements and activate its expression in poplar.

To discover the *cis*-acting elements recognized by PagKNAT2/6b in poplar, we performed a transcription factor-centered yeast one-hybrid (TF-centered Y1H) assay [30]. Sequencing analysis of 144 positive clones revealed that 20 harbored the insertion sequence “GCCTCTT”, which overlapped with the putative nodulin consensus sequence NODCON2GM “CTCTT” when analyzed using the New PLACE webserver. In addition, the Y1H assay confirmed the specific binding of PagKNAT2/6b to the “CTCTT” sequence (Appendix A). Meanwhile, we found all of the key-DEGs containing the “CTCTT” element through screening the 2-kb promoter regions of key-DEGs, and the *PagBOP1/2a* promoter had the highest number of 13 “CTCTT” elements (Appendix A). The results indicated that *PagKNAT2/6b* might directly regulate these leaf development-related genes to affect leaf development in poplar.

We then rescanned the 500-bp *PagBOP1/2a* promoter region and identified 5 “CTCTT” binding sites (Appendix A). To further verify the binding activity of PagKNAT2/6b to “CTCTT”, the 500-bp promoter of *PagBOP1/2a* was re-divided into three fragments P1, P2, and P3, which contain 0, 2, and 3 “CTCTT” binding sites, respectively (Figure 5C). Subsequently, we mutated 2 “CTCTT” sites to “GGGGG” in fragment P2 and used it as a negative control (P2M). Based on the Y1H assay, PagKNAT2/6b can bind to P2 and P3 that contain “CTCTT”, but not P1 and P2M (Figure 5C). Moreover, the EMSA experiment showed that the DNA-Protein complex could be observed when the PagKNAT2/6b protein interacted with the 3× “CTCTT” sequence, and the signal faded gradually by increasing competition for unlabeled probes (Figure 5D). These results indicated that PagKNAT2/6b directly activates the expression of *PagBOP1/2a* through binding to the “CTCTT” element. 

## 3. Discussion

Plants ensure meristem proliferation activities in response to various external stimuli to modify tissue/organs morphology for enhancing plant adaptability [31]. For instance, plants under long-term drought stress will reduce height and leaf size to cope with adverse environments [32]. *KNOX I* genes are involved in maintaining meristems and the specificity of lateral organs [33]. Dominant mutation of the first *KNOX I* gene KN1 of maize alters patterns of development in the leaf [14]. Recently, *AtKNAT3*, a *KNOX II* member, has been identified as a direct regulator in *Arabidopsis* leaf morphogenesis [34], indicating the vital role of *KNOX* family in leaf development. In the perennial poplar, *PagKNAT2/6b* was expressed in leaf primordium, axillary bud, and developing leaves (Figure 1). The altered expression of *PagKNAT2/6b* severely affected the leaf shape of transgenic plants (Figure 2).

We proposed that *PagKNAT2/6b* affects leaf axial development by modulating tissue growth relative to a polarity field. Firstly, *PagKNAT2/6b* was expressed in adaxial, proximal, and medial side cells along the adaxial-abaxial, proximal-distal, and medial-lateral axis, respectively (Figure 1), indicating that it may be involved in axial development. Secondly, altered expression of *PagKNAT2/6b* resulted in changes in the leaf length/width ratio, which is caused by differences in cell differentiation on the proximal-distal and medial-lateral axis (Figure 2G), as well as the asymmetric development of the main and lateral veins on the adaxial and abaxial sides (Figure 3). This indicated that *PagKNAT2/6b* regulated leaf morphology by influencing leaf triaxial development. Moreover, key-DEGs related to leaf development identified from RNA-Seq data support the regulatory role of *PagKNAT2/6b* in leaf development.

To analyze the downstream target genes of *PagKNAT2/6b* in leaf development, we used a TF-centered Y1H assay [30] to identify the *cis*-acting element bound by PagKNAT2/6b. A new “CTCTT” *cis*-element was found as the binding site for PagKNAT2/6b, which is different from the canonical “TGAC” [35]. *PagBOP1/2a*, as the key factor in leaf development, contains 5 “CTCTT” elements in its 500-bp promoter and is highly up-regulated in leaves of *PagKNAT2/6b* overexpression plants (Figure 4B, Appendix A). The leaf phenotypes of *PagKNAT2/6b* overexpression were similar to 35S::*BOP1* and 35S::*BOP2 Arabidopsis*. In addition, *BOP1* and *BOP2* are relatively upstream transcription factors in establishing leaf polarity through regulating *PHB*, *FIL*, and *AS2* [25,29]. Therefore, to confirm the potential activation of PagKNAT2/6b on the expression *PagBOP1/2a*, we have performed transient expression and EMSA assay (Figure 5A,D), which showed that PagKNAT2/6b could up-regulate *PagBOP1/2a* through binding to its “CTCTT” *cis*-element in the promoter region. Taken together, we concluded that PagKNAT2/6b could directly regulate *PagBOP1/2a* to modulate leaf polarity development.

The first confirmed *KNOX I* gene conserved binding *cis*-acting element is “TGAC” reported in barley [36]. Then, it has been shown that *KNOX I* gene KN1 in maize can bind to the “TGAC” elements in the promoter regions of *HB*, *YABBY*, and *REVOLUTA* (*REV*), and these genes may coordinate the development of leaf polarity [22]. In this study, we identified that PagKNAT2/6b binds to the NODCON2GM element “CTCTT” of *PagBOP1/2a* in poplar leaf development, instead of the reported binding sequence “TGAC” of gibberellin catabolism gene *ga20x1* by *KN1* in the establishment and maintenance of SAM in maize [35]. The NODCON2GM element bonded by PagKNAT2/6b, as a nodulin consensus sequence, is conserved among nodulin genes [37,38]. Interestingly, *MtKNAT3*, *MtKNAT4,* and *MtKNAT5* (*MtKNAT3/4/5*) are highly expressed during nodule formation, and *MtKNAT3*, an orthologue of the *Arabidopsis*
*KNAT3**,* overexpression could induce the formation of root-nodules in *Medicago truncatula* [39]. In addition, *MtEFD1* (Ethylene response Factor required for nodule Differentiation) is regulated by MtKNAT3/4/5 during nodule development [40]. Two “CTCTT” elements were found in the 500-bp promoter region of *MtEFD1* (Appendix A). These results suggested that KNOX might regulate the nodulin genes through “CTCTT” elements to promote nodulation in *M. truncatula*. Therefore, in contrast to the previous suggestion that KN1 regulates the expression of *ga2ox1* by binding the “TGAC” element in maize, it was shown that their members may have evolved to different transcriptional regulation modes in different scenarios across tissues/species.

Here, we proposed PagKNAT2/6b-PagBOP1/2a regulatory module (Figure 6). When drought signals stimulate the expression of *PagKNAT2/6b* in poplar [27], the high abundance of PagKNAT2/6b directly activates the expression of *PagBOP1/2a*. The latter changes the rate of axial differentiation of cells, thereby regulating the leaves developed in a small and wrinkled manner. Size-reduced and curled leaves can effectively reduce water evaporation, thus increasing drought resistance.

## 4. Materials and Methods

### 4.1. Plant Materials

Hybrid poplar (*Populus alba × P. glandulosa*) ‘84K’ and the *35S:**PagKNAT2/6b*, *PagKNAT2/6b*-SRDX, and p*PagKNAT2/6b::GUS* transgenic plants were obtained previously [21]. One or two-month-old soil-cultured poplar materials were grown at 23–25 °C under a 16/8 h (day/night) photoperiod. Leaf plastochron index (LPI) was used to index the age of leaves [28].

### 4.2. Leaf Morphological Analysis

The leaf of non-transgenic controls (84K) and *PagKNAT2/6b* transgenic lines were detached and photographed. The leaves were spread and fixed in 5% agarose and the length and width of the leaves were calculated using ImageJ software. Subsequently, the ratio of length/width was calculated.

### 4.3. Tissue Sections Preparation and Microscopy

GUS staining was conducted as previously reported [41]. In brief, one-month-old *P_PagKNAT2/6b_::GUS* transgenic plants were incubated in 90% acetone for at least 2 h at ice, and washed three times with wash buffer (0.2 mol/L NaH_2_PO_4_, 0.2 mol/L Na_2_HPO_4_, 2 mmol/L K_3_[Fe(CN)_6_], 2 mmol/L K_4_[Fe(CN)_6_]) and then set the plants in GUS stain buffer (wash buffer with 1 mM X-Gluc) at 37 °C for 10 h. Following staining, plants were cleared with 75% ethanol. A series of leaves from *P_PagKNAT2/6b_::GUS* transgenic plants were sectioned using a vibratome (Leica). The SAM, LPI1, and LPI2 of *P_PagKNAT2/6b_::GUS* transgenic lines were cut into a 30-μm-thick section using vibratome and stained with 0.05% (*w*/*v*) toluidine blue O for 1 min. Finally, the sections were observed with bright light under Zeiss Axio Scope (Zeiss, Docuval, Germany).

### 4.4. Quantitative Real-Time PCR

RNA was isolated using an RNeasy Plant Mini Kit following the manufacturer’s protocols (Qiagen, Hilden, Germany). cDNA synthesis was carried out with 1 μg RNA using the Superscript III reverse transcription kit (Thermo Fisher, In. Co. Ltd., Waltham, MA, USA). qRT-PCR was performed using the SYBR Premix ExTaqTM Kit (TaKaRa, Beijing, China) on a Roche Light Cycle 480 Real-time PCR System (Roche, Basel, Switzerland). At least six seedlings were pooled for each sample and at least three biological replicates were examined. Relative expression levels were normalized to the expression of *PtACTIN* (Potri.001G309500) and *PtUBIQUITIN* (Potri.001G418500). The gene-specific primers for qRT-PCR are listed in Appendix A.

### 4.5. RNA-Seq and Data Analyses

LPI2 to LPI4 leaves were collected separately from two-month-old control, *PagKNAT2/6b* overexpression and *PagKNAT2/6b*-SRDX plants, and total RNAs were isolated using the RNeasy Plant Mini Kit (Qiagen, Hilden, Germany), three biological replicates for each genotype. Approximately 20 μg of total RNA was used for Illumina sequencing at GENE DENOVO (Guangzhou, China). Each library was constructed by NEBNext Ultra RNA Library Prep Kit for Illumina (NEB, Beijing, China). Fragment per kb per million reads (FPKM) was used to estimate the abundance of the transcripts. The DESeq2 package (http://www.r-project.org/) was used to identify differentially expressed genes (DEGs) across genotypes. We identified genes with |log2FC|>1 and a false discovery rate (FDR) corrected *p*
< 0.05 as significant DEGs. The functional analysis of DEGs was performed using Gene Ontology (GO) enrichment through AgriGo [42]. The sequencing reads from each sequencing library have been deposited at National Center for Biotechnology Information (NCBI) with the Project ID PRJNA719233.

### 4.6. Co-Expression Analysis

For co-expression network construction, expression data were obtained in the *Populus* Gene Atlas database (https://phytozome.jgi.doe.gov) which covers various tissues and organs. Pearson correlation coefficients (PCCs) were calculated in parallel between all pairs of gene expression vectors. A threshold of *p*
< 0.05 and absolute PCC ≥ 0.90 were applied to the significant correlations, and their co-expression relationships were visualized by Cytoscape [43].

### 4.7. Transient Expression Assay

Transient expression assays in *Nicotiana benthamiana* leaves were performed as previously described [44]. The 2-kb promoter of *PagBOP1/2a* was amplified and cloned into pGreenII 0800-LUC vectors, which generated the reporter constructs *P_PagBOP1/2a_::LUC*, respectively. The effector (*35S::**PagKNAT2/6b*) was constructed by cloning the open reading frame of the *PagKNAT2/6b* into the pGreenII 62-SK vector. The recombinant plasmids and negative control vectors were introduced into *Agrobacterium tumefaciens* GV3101. The effector and reporter vectors were then co-transformed into *N. benthamiana* leaves as previously described [44]. GloMax 20/20 Luminometer (Promega, Madison, WI, USA) and a Dual-Luciferase Assay Kit (Promega, Madison, WI, USA) were used to test the LUC activity. The experiment was performed more than three times and included three biological replicates for each experiment.

### 4.8. Yeast One-Hybrid Assay (Y1H)

Yeast one-hybrid assay was performed as previously reported [45]. The assay was carried out using the Matchmaker™ Gold Yeast one-hybrid System (Clontech, Beijing, China). Different promoter fragments from the *PagBOP1/2a* promoter were inserted into the pHIS2 vector as the reporter. The coding sequence (CDS) of the *PagKNAT2/6b* gene was inserted into the pGADT7-Rec2 vector to generate the effector. Then, the effector and reporter constructs were co-transformed into the Y187 yeast strain, and the transformed yeast cells were selected and screened on SD medium lacking leucine (Leu) and tryptophan (Trp) (SD/–Leu/–Trp, DDO) and on SD medium lacking Leu, Trp, and histidine (His) (SD/–Leu/–Trp/–His, TDO) medium with 40 mM 3-AT (3-Amino-1, 2, 4-triazole). p53HIS2 + pGAD-Rec2-53 and pHIS2 + pGAD-Rec2-PagKNAT2/6b were used here as positive controls and negative control, respectively. The experiment was performed three times and similar results were obtained.

### 4.9. Transcription Factor (TF)-Centered Y1H Analysis

TF-Centered Y1H screening was performed to determine the *cis*-acting elements bound by PagKNAT2/6b according to Ji et al. [30], and the Y187 yeast library containing random short DNA sequences insertion of pHIS2 was kindly provided by Prof. Yucheng Wang (Shenyang Agricultural University) as baits. The CDS of *PagKNAT2/6b* was cloned into vector PGADT7-Rec2 as the prey. In brief, 2 mL tube containing Y187 yeast library cells, 1 μg PGADT7-PagKNAT2/6b, 25 μL carrier DNA (stock concentration 2 mg/ml), 17 μL LiAc (stock concentration 1 M), and 120 μL PEG3350 (stock concentration 50%) were incubated at 42 °C for 2 h. The yeast cells were pelleted by centrifuge at 700× *g* for 5 min at room temperature, then the supernatant was removed, and the pellet was re-suspended in 100 μL water. The cells were spread on DDO and TDO containing 40 mM 3-AT to select the transformants. p53HIS2 + pGAD-Rec2-53 and Y187 yeast library + pGADT7-Rec2 were used here as positive controls and negative, respectively. All positive clones were sequenced after amplification with F and R primers of pHIS2 (Appendix A), and the insertion sequences were analyzed to determine *cis*-acting elements using New PLACE (https://www.dna.affrc.go.jp/PLACE, 5 March 2022). Finally, the selected element was validated with Y1H as described above.

### 4.10. Electrophoretic Mobility Shift Assay (EMSA)

The His-PagKNAT2/6b recombinant proteins were prepared according to our previous study [21] and purified with beads IDA-Nickel (Sloarbio, Beijing, China). The probes with or without biotin labeling were synthesized by the company (Tsingke, Beijing, China). The probes were then incubated with the recombinant proteins and assayed using the EMSA kit (Beyotime, Shanghai, China). The sequence of the probe used for EMSA is listed in Appendix A.

### 4.11. Statistical Analysis

Data are presented as means ± standard deviations (SD). The significance of differences was calculated using the Student’s *t*-test. Differences were considered significant when *p*
< 0.01. At least three biological replicates were used in all tests.

## 5. Conclusions

In conclusion, PagKNAT2/6b directly activates the expression of *PagBOP1/2a* under drought stress to change leaf shape, which confirmed our previous hypothesis that PagKNAT2/6b acts as an upstream key transcription factor influencing leaf morphogenesis by regulating the expression of downstream leaf development-related genes. Therefore, PagKNAT2/6b-PagBOP1/2a regulatory module provides a strategy for the molecular breeding of woody plants with drought tolerance by adjusting leaf architecture.

## Figures and Tables

**Figure 1 ijms-23-05581-f001:**
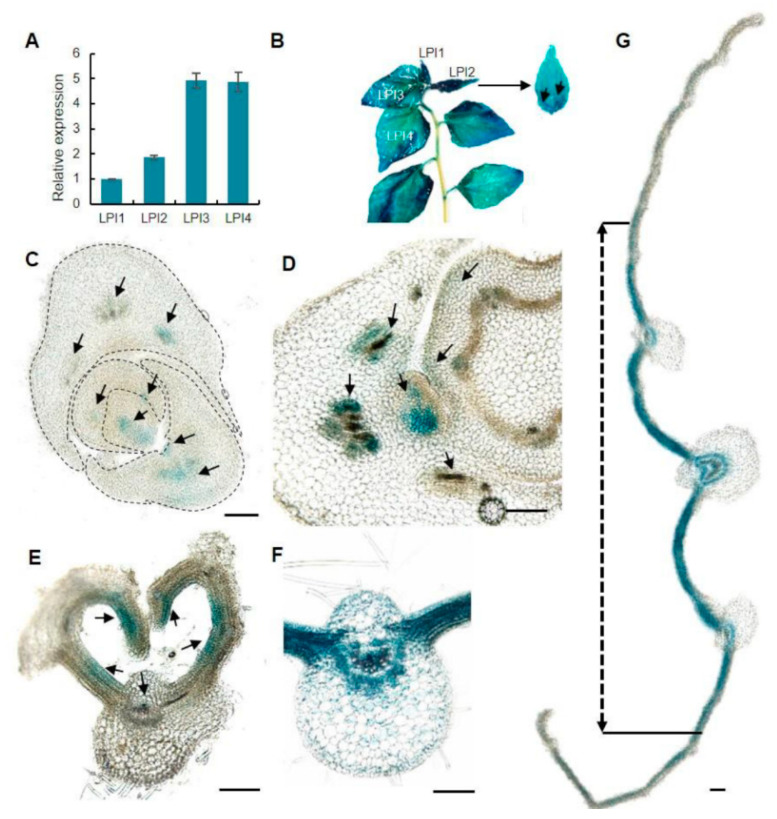
Expression patterns of *PagKNAT2/6b* in poplar leaf. (**A**) The expression profile of *PagKNAT2/6b* during leaf development from leaf plastochron index 1 (LPI1) to LPI4 by qRT-PCR. (**B**) GUS staining of *P_PagKNAT2/6b_*::GUS transgenic poplar in LPI1 to LPI4, and LPI2 in a magnified view. (**C**,**D**) Transverse sections of the shoot apical meristem (SAM). (**E**) GUS signals in unexpanded leaf LPI1. (**F**) GUS signals in the main veins of LPI2. (**G**) GUS signals in the middle of LPI3, and the arrow shows the medial region of LPI3. Experiments were repeated three times and the representatives were shown. The salient signal is indicated with arrows. Bar = 100 μm.

**Figure 2 ijms-23-05581-f002:**
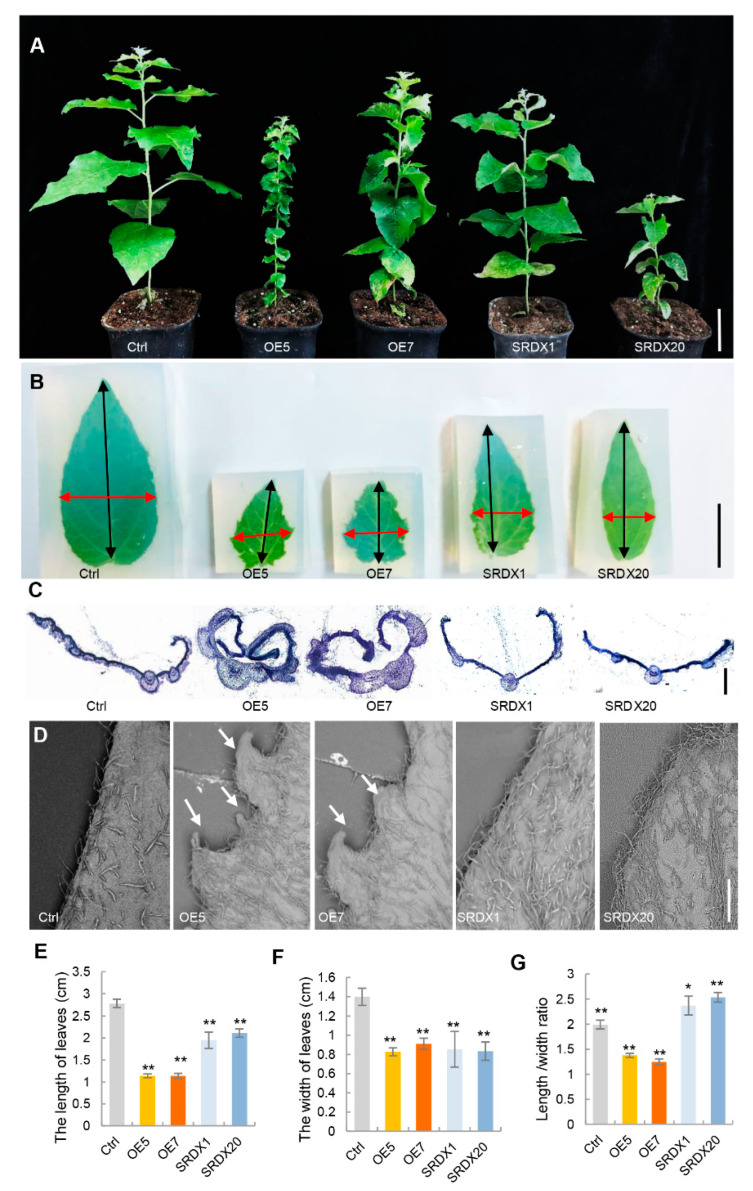
Morphological comparisons of *PagKNAT2/6b* transgenic and non-transgenic control poplar. (**A**) Phenotypes of the seedlings of non-transgenic control poplar (Ctrl), *35S:PagKNAT2/6b* overexpression lines (OE5 and OE7), and *PagKNAT2/6b*-SRDX dominant repression lines (SRDX1 and SRDX20). Bars = 6 cm. (**B**) Phenotypes of LPI3 leaves of Ctrl and *PagKNAT2/6b* transgenic plants, respectively. The leaves were embedded in agarose and spread out to facilitate the measurement of leaf length and width. Bars = 1 cm. (**C**) The section overviews of the red horizontal arrow area in (**B**) of LPI3. Bars = 100 μm. (**D**) The leaf margin morphology was shown by scanning electron microscopy. Bars = 1 mm. (**E**–**G**) The compute of leaf length, width, and length/width ratio of LPI3 in control and transgenic plants, respectively. Asterisks denote significant differences compared to Ctrl, * *p* < 0.05 or ** *p* < 0.01.

**Figure 3 ijms-23-05581-f003:**
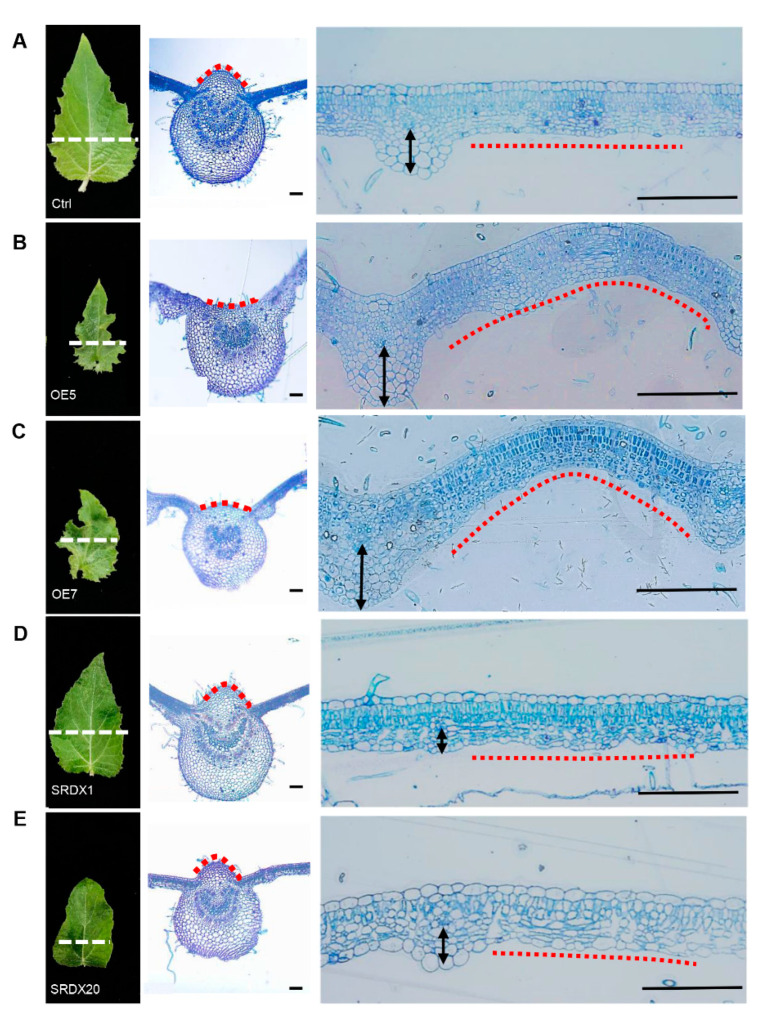
Abnormal expression of *PagKNAT2/6b* affects leaf adaxial-abaxial development. The left panel of (**A**–**E**) are morphologic overviews of LPI3. The white dotted line shows the positions for microsection preparation in the middle and right panels. The middle panel shows views of the main leaf midrib, and the red dotted lines point to the differences on the adaxial side of the main leaf midrib in different genotypes (**A**–**E**). The right panel shows views of the lateral leaf midrib, and the black arrows and red dotted lines show the difference on the abaxial side of the lateral leaf midrib of different genotypes (**A**–**E**). (**A**–**E**), Ctrl, OE5, OE7, SRDX1, and SRDX20, respectively. Bars = 100 μm.

**Figure 4 ijms-23-05581-f004:**
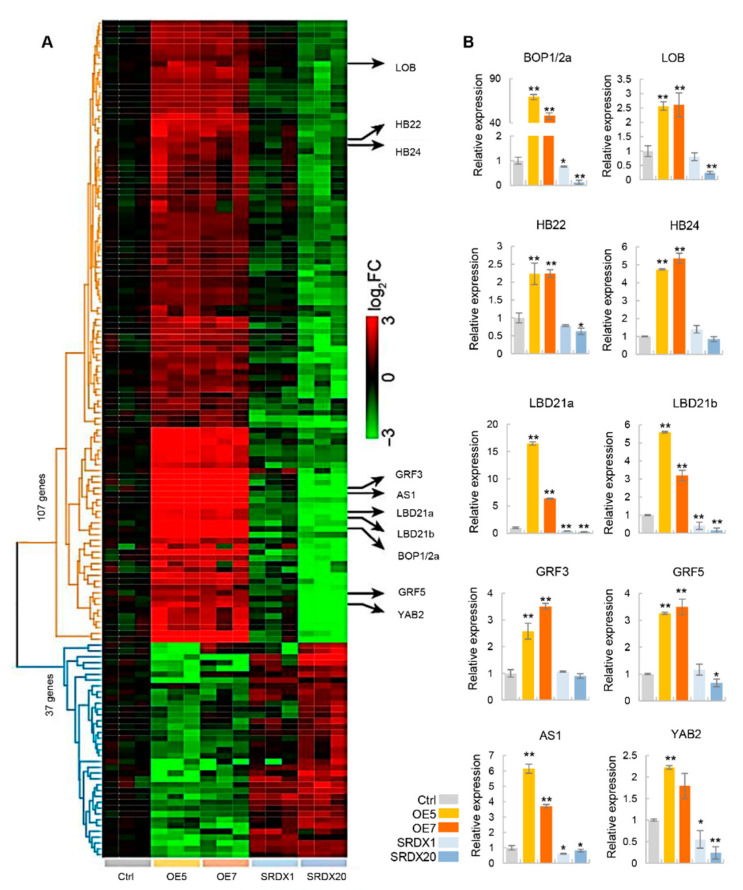
The analysis of changes in gene expression in leaves in *PagKNAT2/6b* transgenic lines. (**A**) Heatmap of common differentially expressed genes (common-DEGs) in control plants (Ctrl) and *PagKNAT2/6b* transgenic lines. A total of 107 genes were up-regulated in OE5 or OE7 and down-regulated in SRDX1 or SRDX20, and 37 genes were down-regulated in OE and up-regulated in SRDX lines. The ten key-DEGs related to leaf development are marked on the right. (**B**) qRT-PCR validation of the relative expression of the key-DEGs. Asterisks denote significant differences compared to Ctrl, * *p* < 0.05 or ** *p* < 0.01.

**Figure 5 ijms-23-05581-f005:**
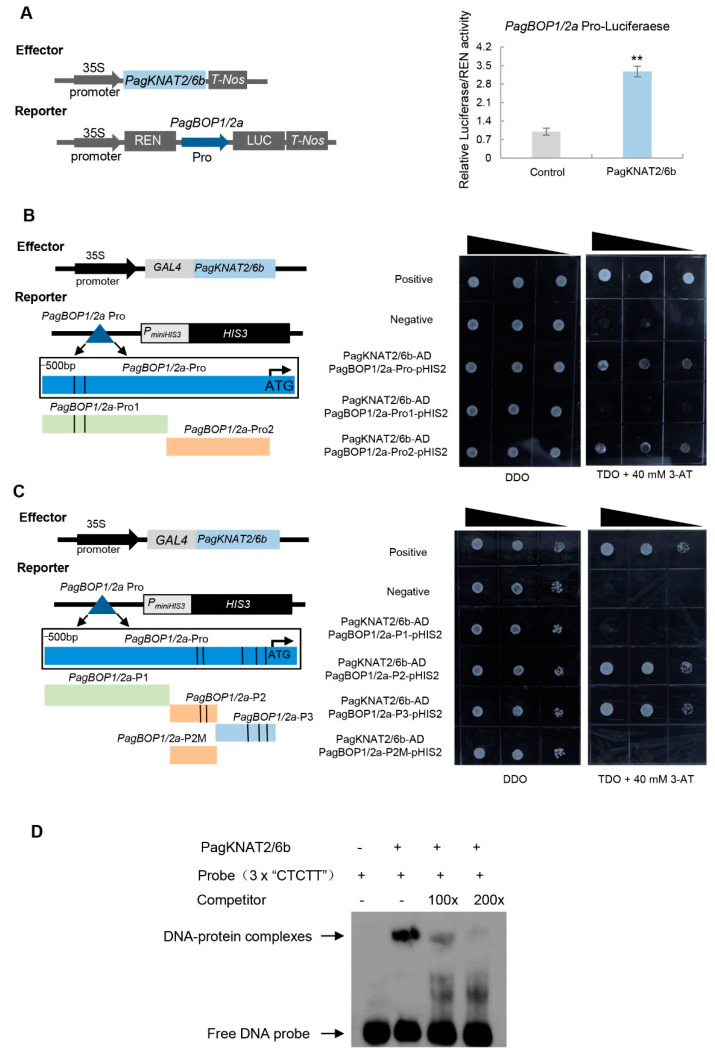
The regulation of *PagBOP1/2a* by PagKNAT2/6b through the “CTCTT” element. (**A**) Transient transcriptional activity assay to examine the regulation of *PagBOP1/2a* by PagKNAT2/6b. The 2-kb promoter region of *PagBOP1/2a* was used to construct the reporter vector. Asterisks represent significant differences compared to control, ** *p* < 0.01. (**B**) Yeast one-hybrid assay to examine binding affinity of PagKNAT2/6b for the different fragments including “TGAC” elements or not of the promoter of *PagBOP1/2a*. *PagBOP1/2a*-Pro (500-bp promoter region), *PagBOP1/2a*-Pro1 (including two “TGAC” elements, the black lines), and *PagBOP1/2a*-Pro2 (without “TGAC” elements) were inserted separately into vector pHIS2 and then co-transformed into Y187 with PagKNAT2/6b-AD (pGAD-Rec2-PagKNAT2/6b), respectively. (**C**) Yeast one-hybrid assay to examine binding affinity of PagKNAT2/6b for the different fragments including “CTCTT” elements or not of the promoter of *PagBOP1/2a*. *PagBOP1/2a*-P1 (without “CTCTT” elements), *PagBOP1/2a*-P2 (including two “CTCTT” elements, the black lines), *PagBOP1/2a*-P3 (including three “CTCTT” elements), and *PagBOP1/2a*-P2M (the two “CTCTT” elements in fragment P2 were mutated to “GGGGG”) were separately inserted into vector pHIS2 and then co-transformed into Y187 with pGAD-Rec2-PagKNAT2/6b (PagKNAT2/6b-AD), respectively. Vector combinations p53HIS2 + pGAD-Rec2-53 and p53HIS2 + pGAD-Rec2-PagKNAT2/6b were used as the positive and negative control, respectively, in B and C. (**D**) Electrophoretic mobility shift assay (EMSA) for detection of PagKNAT2/6b binding to “CTCTT” sequence. EMSA was performed using the purified PagKNAT2/6b proteins and 3× “CTCTT” sequence labeled with biotin as probes. Competition to the labeled probe was tested by adding 100- and 200-fold excess of the unlabeled probes.

**Figure 6 ijms-23-05581-f006:**
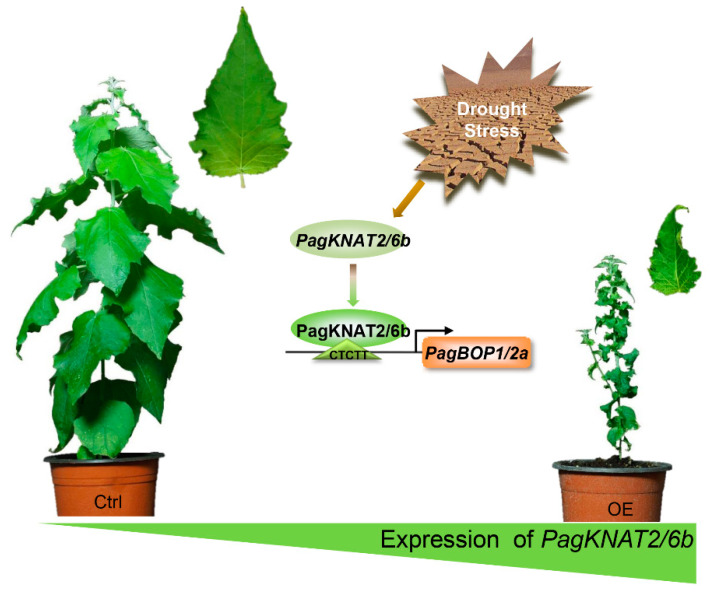
The PagKNAT2/6b-PagBOP1/2a regulatory module regulates leaf morphology under drought stress. The leaves receive drought signals and produce more PagKNAT2/6b proteins, which bind to the “CTCTT” element of the *PagBOP1/2a* promoter and activate the expression of *PagBOP1/2a* for the regulation of leaf development.

## Data Availability

The original data presented in this study are included in the study, and the sequencing reads from each sequencing library of CK, PagKNAT2/6b OE, and SRDX leaf have been deposited at NCBI with the Project ID PRJNA719233. This data can be found here: [https://www.ncbi.nlm.nih.gov/bioproject/PRJNA719233 (accessed on 11 May 2022)].

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
