# Peer review of "The PagKNAT2/6b-PagBOP1/2a Regulatory Module Controls Leaf Morphogenesis in Populus"

_ijms, 2022, doi:10.3390/ijms23105581_

Round 1
Reviewer 1 Report
The manuscript “The PagKNAT2/6b-PagBOP1/2a regulatory module controls leaf morphogenesis in Populus” by Zhao et al is a detailed functional characterization of the factor involved in the leaf morphogenesis. The experiments are conducted in a good manner and the data presented in the manuscript is organized well which provides enough evidence to prove their research question. In my opinion, the manuscript meets the standards of IJMS and should be accepted for publication. Please consider some little corrections which I mentioned below.
Line 21-26: Please rephrase this long sentence and break it down into smaller sentences.
Line 32: Check with italic
Line 77-78: Check with italic
Line 85: Please rephrase
Line 87: Please explain LPI3-4 when it comes the first time in the text
Reviewer 2 Report
- Lack of the hypothesis.
- In Introduction were given Conclusions!
- GUS was not introduced.
- Problems with times of verbs, confusion between published and exposed as results.
- Age, size, form of plants???
- Statistics could be shown at the end of M&M.
- Lack of Final Remarks and Conclusions. Conclusions must respond to hypothesis and underline the novelty of the study. Conclusions cannot content the references, tables, figures or discussion.
- Problems in phrases that are too long, or without verbs, or repeating words, or lacking words. Sometimes difficult to understand, or impossible. Please use verbs in pass to talk about your results, and in present to express all published.
- Detailed observations you will find in pdf attached to revision.

Reviewer 3 Report
The paper entitled “The PagKANT2/6b-PagBOP1/2a regulatory module controls leaf morphogenesis in Populous” by Yan-Qiu Zhao described the functional characterization of KNAT2 and BOP in leaf morphogenesis in Poplar. KNAT2 is on of the important transcription factors for plant development, including leaf morphogenesis.
The author successfully developed OE and SRDX lines for KNAT2/6b and characterized the phenotype of the lines, and found that the gene is involved in regulating the development of leaves in both the proximal-distal axis and medial-lateral axis. They also performed RNA-seq analysis for OE lines and identified down-stream regulators. They showed that BOP1/2 is directly regulated by KNAT2. in Poplar
The results of this paper would provide a useful information and would attract wide range of readers. The experiments and analyses are carefully performed and the conclusions are reasonable. I do not see any big problem on this paper. As a conclusion, the paper would be sufficient to merit publication in IJMS though a minor revision is recommended which needs to include the following points.
(1) Line 126: I do not agree with this. Both of the leaf aspect ratio and margin appear to be equally affected.
(2) It may be useful to investigate whether the expression of BOP changes with drought stress to reinforce the hypothesis in Figure 6.
Reviewer 4 Report
I want to thank the authors for an interesting study. The manuscript interested me very much. I will recommend the manuscript for publication.
However, I have a few comments:
- The text contains many gene names and they are written in different styles. It would be nice if the authors would unify the names of the genes they use.
- How does the ‘84K’ hybrid reproduce? Does it go through an in vitro stage, like transgenic plants?
- It would be appropriate if Table S4 included the primer annealing temperatures and the lengths of the amplified fragments (at least in those cases where this information can be demonstrated).
